



# Trade wind regimes during the Great Barrier Reef coral bleaching season

Lara S. Richards[1,2], Steven T. Siems[1], Yi Huang[2,3], Daniel P. Harrison[4], and Wenhui Zhao[3]

[1]Monash University
[2]The ARC Centre of Excellence for 21st Century Weather
[3]The University of Melbourne
[4]National Marine Science Centre, Southern Cross University

**Correspondence:** Lara S. Richards (Lara.Richards@monash.edu)

**Abstract.** The trade winds over the Great Barrier Reef (GBR) dominate the local weather in the region, bringing cooler and drier air over the Reef, which promotes ocean cooling. The absence of the trade winds is often marked by periods of weaker winds and higher humidity, known as the doldrums, which cause ocean temperatures to spike and can develop into marine heatwaves that lead to coral bleaching. As the shallow waters of the GBR are strongly tied to the local meteorology,
studying the evolution and structure of the trade winds during the austral warmer months is essential for understanding the development of thermal bleaching events. Through a K-Means cluster analysis on reanalysis soundings from Davies Reef from December-April 1996-2024, we find the formation of the doldrums is linked to the passage of a Rossby-wave train over eastern Australia. Years with mass thermal bleaching are correlated with more doldrums days, but also less of the strong trade wind days in December and April which can promote early-summer warming and allow warmer temperatures to persist later into
the season.

## 1 Introduction

Covering almost half of the Earth's surface, the trade winds carry colder air from higher latitudes into the tropics, while their strength and location vary annually following the position of the inter-tropical convergence zone as it migrates with the seasons (Crowe, 1950; Malkus, 1958; Wyrtki and Meyers, 1976). The trades play an essential role in regulating subtropical ocean
temperatures (Liu and Philander, 1995; Merrifield, 2011; Takahashi and Watanabe, 2016), typically cooling them as the strong surface winds enhance the latent heat flux (LHF) promoting evaporational cooling and ocean mixing (Skirving et al., 2006). The trades also enhance surface convection, as the strong winds promote the formation of trade cumulus (Malkus, 1958; Nuijens and Stevens, 2012) which reduces the amount of short-wave flux (SWF) reaching the ocean's surface.

While the trade winds are typically sustained in the winter hemisphere (Wyrtki and Meyers, 1976), they can frequently weaken or collapse in the summer hemisphere as tropical disturbances, such as monsoon troughs and tropical cyclones/storms, progress through the region (Holland, 1986; Pope et al., 2009; Murphy et al., 2016). As the presence of the trade winds promotes ocean cooling, when they weaken or collapse local ocean temperatures can spike. This collapse can result in what's known as the



doldrums, creating areas of weak winds and high humidity which dampen the LHF (Karnauskas, 2020; Richards et al., 2024).
At the same time, the lack of boundary layer turbulence limits cloud formation allowing more SWF into the ocean promoting
ocean heating. While the doldrums have more explicitly been studied over the tropical Atlantic (Klocke et al., 2017; Wind-
miller, 2024), reports of 'calm and clear' conditions appearing in trade wind regions can be traced back to the 1950's (Crowe,
1950) and are a common feature of thermal coral bleaching studies (Glynn, 1968; Smith, 2001; Jokiel and Brown, 2004; Bain-
bridge, 2017; Baird et al., 2018; Richards et al., 2024).


The majority of the world's coral reefs lie in trade wind zones, including the Great Barrier Reef (GBR), located in the Coral
Sea on the Australian eastern coast, which hosts the world's largest coral reef ecosystem of tremendous biological, cultural,
and economic value. One of the many pressures facing the Reef is the increasing frequency of mass coral bleaching events
(CBEs), with nine recorded since 1998, six of which occurring since 2016 (AIMS, 2025). These events are increasing due to
the background warming of ocean temperatures from climate change (Henley et al., 2024). The shallow waters of the Great
Barrier Reef are particularly sensitive to changes in weather patterns and cloud cover (Harrison et al., 2019; Zhao et al., 2021),
where shifts from the trade winds to doldrums conditions can cause ocean temperature spikes and potentially coral bleach-
ing (Karnauskas, 2020; Richards et al., 2024). Connections between the local and synoptic meteorology were explored by
Richards et al. (2024), who followed the local and synoptic meteorological evolution at Davies Reef during the 2022 CBE.
The initial spike in ocean temperatures was linked to the weakening and eventual breakdown of the trade winds. As the trade
winds collapsed, wind speeds decreased and surface humidity rose, dampening the LHF. Following the event's peak, ocean
temperatures rapidly fell as the trade winds re-established bringing cooler, drier air and stronger surface winds over the site,
allowing the LHF to triple. The collapse and re-establishment of the trades can be traced to the extra-tropics, where a string
of Rossby-wave breaking events and continual cut-off low formation was thought to initiate the trade's breakdown, while the
trade's re-establishment was attributed to a strong coastal ridging event that surged up the Australian east coast.

While other synoptic typing studies have provided valuable insights into the northern Australian wet season (Lyons and Bonell,
1992; Pope et al., 2009), a climatology of GBR weather patterns and a more detailed understanding of the trade-wind 'cycle'
during thermal bleaching events is missing from the literature. Thus, we aim to characterise the trade wind regimes over the
GBR during the Australian coral bleaching season (December-April) and understand their influence on GBR ocean tempera-
tures. Using K-Means clustering of reanalysis data on a central GBR site, we construct our clusters from the thermodynamics
of the boundary layer and trade-wind layer of the lower free troposphere. Designating our clusters into a trade wind and non-
trade wind regime, we use our clusters to show the synoptic differences between bleaching and non-bleaching years and their
evolution throughout recent bleaching events.





## 2 Data and Methodology

### 2.1 Cluster analysis

To categorise the dominant weather patterns during the GBR coral bleaching season, K-Means clustering is performed on atmospheric profiles at Davies Reef (Fig. 1 first column) to determine regional weather types based on the thermodynamic structure of the boundary layer and middle atmosphere where the trade wind layers persist. As the GBR covers a large latitudinal range (10-24° S), Davies Reef (18.83° S, 147.63° E) is chosen for the clustering location as it is both in the central GBR, with more frequent CBEs recorded, and has a highly consistent observational record of both meteorological and ocean parameters through the Australian Institute of Marine Science (AIMS) Davies Reef Automatic Weather Station (AWS). Due to an absence of observational soundings over the GBR, atmospheric reanalysis data from the European Centre for Medium-Range Weather Forecasts' fifth reanalysis (ERA5) (Hersbach et al., 2020), produced on hourly intervals at 0.25° resolution, taken at the closest grid point to Davies Reef (18.75° S, 147.75° E) is used to construct the clusters. Air temperature, dew point temperature, and the horizontal wind components are taken at the surface, 925 hPa, 850 hPa, 700 hPa, 600 hPa, and 500 hPa, using only the 0000 UTC (1000 LST) time step to give a daily snapshot of the atmosphere from the surface to the mid-troposphere, well above the trade inversion. The profiles are restricted to 500 hPa to better capture the diversity in the boundary layer and trade wind structure. Despite the large latitudinal extent of the GBR, the clusters produced at Davies Reef was found to capture the diversity of the northern and southern sectors. A clustering analysis was performed at Lizard Island and Heron Island (not shown), where similar weather regime clusters are found.

To capture the recent GBR CBEs in the clustering analysis, the Davies Reef ERA5 data were extended to cover the period from December-April for the years of 1996-2024. As this manuscript was prepared before the 2025 CBE was declared, we only include the eight events from 1998 to 2024. While the GBR coral bleaching season was originally defined as January-April by (Zhao et al., 2021), this study expands the definition to include December in order to fully capture the austral summer.

### 2.2 Synoptic analysis

After each day in the clustering period is assigned, the average surface synoptic conditions are calculated for each cluster followed by their anomalies. The synoptic analysis combines ERA5 mean sea-level pressure and 10 m horizontal winds with the satellite derived National Oceanic and Atmospheric Administration's (NOAA) Daily Optimum Interpolation Sea Surface Temperature (DOISST) version 2.1 (Huang et al., 2021). Day-of-year based anomalies are calculated using a daily mean climatology for the 1996-2024 period to remove the cluster's seasonal biases. The sea surface temperature (SST) anomalies have been linearly detrended to remove the long-term warming signal and have then undergone a simple one sided t-test where only the anomalies with a p-value < 0.01 are considered. Lastly, to understand the origin of the boundary layer air masses, back trajectories are produced for each cluster using the Lagrangian analysis tool Lagranto version 2.0 (Sprenger and Wernli, 2015), which uses ERA5 data on 6-hourly intervals. The daily back trajectories each start at 0000 UTC running for 72 hours from Davies Reef at 925 hPa.





### 2.3 Davies Reef observations

Daily averages of atmospheric and oceanic observations from the Davies Reef AWS are used throughout the clustering period,
90  noting that observations from April 23$^{rd}$-30$^{th}$ 2024 were unavailable and therefore omitted from all analysis. The Davies Reef
daily averaged 4 m ocean temperature ($T_{4m}$) is considered our primary ocean temperature record as many other AIMS GBR
AWS sites contain frequent missing observations. For rare occasions when the Davies Reef $T_{4m}$ is not available, missing data is
filled by interpolating the $T_{2m}$ and $T_{8m}$ records (0.59% of data). When this interpolation over depth was not possible, the daily
$T_{4m}$ was estimated by averaging the daily $T_{4m}$ from the nearest available days (0.07% of data). For periods when the Davies
95  Reef AWS wind and air temperature observations were not available, missing data was filled using ERA5 data (2.06% of data)
or by averaging the nearest available days (0.08%). We note ERA5 was previously found to have a high correlation with the
AIMS Davies Reef observations(Richards et al., 2024). As the rain accumulation and relative humidity are only available from
2008 onwards, no gap filling was performed and the total cluster days assigned is stated Table 2.

Due to the lack of in-situ fluxes, a local net surface energy budget is produced from ERA5 at Davies Reef using daily av-
eraged mean net surface fluxes. Here we analyse the contribution of the radiative fluxes (short-wave ($Q_{SW}$) and longwave
($Q_{LW}$)) and the turbulent fluxes (latent heat ($Q_E$) and sensible heat ($Q_H$)) to the total net flux ($Q^*$) (Eq 1).

$$Q^* = Q_{SW} + Q_{LW} + Q_H + Q_E. \tag{1}$$

### 2.4 Cloud cover

The satellite imagery from the Japan Aerospace Exploration Agency's (JAXA) Himawari-8/9 satellite are used to investigate
the cloud fields that are associated with clusters at Davies Reef. Himawari-8/9 uses the International Satellite Cloud Clima-
tology Project (ISCCP) cloud classification system which uses cloud top pressure and cloud optical thickness to determine
the cloud type (Nakajima and Nakajima, 1995; Kawamoto et al., 2001). The cloud types are sorted into low clouds (cumulus,
stratocumulus, and stratus), mid-level clouds (altocumulus, altostratus, and nimbostratus), and high clouds (cirrus, cirrostratus,
and deep convection). It should be noted that Himawari-8/9 is a passive sensor, meaning it cannot detect multi-layer clouds,
low cloud may be underrepresented when upper-level clouds are optically thick (Marchand and Ackerman, 2010), thus higher
clouds may be overrepresented in cloud fraction calculations. Here we consider only a 1° box centred on Davies Reef tak-
ing only the 0000 UTC observations. Note that this analysis is limited to time period of December 2015 to April 2024, as
Himawari-8/9 was launched in 2015.

### 2.5 MJO indices

Lastly, as the Madden-Julien Oscillation (MJO) is known to influence tropical winds and convection in the Australian tropics,
and potential links have been noted between the MJO and Australian marine heatwaves (Holbrook et al., 2019), we analyse
the phase distribution between our clusters. Here we utilise the Bureau of Meteorology's Real-time Multivariate MJO index
(Wheeler and Hendon, 2004) to analyse the MJO phase and amplitude and how they differ between our clusters.





## 3 Cluster climatology

### 3.1 Cluster identification


In the cluster analysis at Davies Reef, five clusters were found to be the optimal number to best represent the diversity in the atmospheric profiles. To help characterise each cluster, composite soundings from the surface to the mid-troposphere were produced (Fig. 1) with their monthly frequency (Fig. 2). The five clusters are split into three 'trade wind' clusters (classic trades, summer trades, and wet trades) and two 'non-trade wind' clusters (doldrums and northerlies) and are ordered based

on their heading and wind speed. The classic trades display a common trade wind structure, where surface south-easterlies weaken and turn to westerlies at upper levels (Fig. 1a). The classic trades are scarce during December-March however become the dominant cluster (∼60%) in April (Fig. 2) as the Australian monsoon retreats and the subtropical ridge moves north (Fig. 1b). In the boundary layer, the thermodynamic profile of the summer trades compares to the classic trades, whilst above it, the summer trades lack an upper-level wind shift also becoming warmer and drier aloft (Fig. 1d). The omega profiles are also

similar between the classic and summer trades, with only slightly weaker surface ascent separating the summer trades. The summer trades are evenly distributed across December-April, averaging around 20% frequency in each month. The wet trades differ significantly from the other trade clusters, with a warmer and more moist profile and a larger easterly component to the lower-level winds (Fig. 1g). The wet trades omega profile follows a similar shape to the other trade wind clusters, although with significantly stronger surface ascent. As the wet trades are more frequent between January-March (∼30%), when the monsoon

is active, the higher atmospheric moisture is not surprising. Moving to the non-trades clusters, the doldrums are characterised by their weak lower-level winds and strong boundary layer temperature inversion (Fig. 1j). Along with having comparatively weaker ascent near the surface, above this inversion the air is typically descending. The doldrums are most frequent between December-February, becoming scarce by April. Lastly, the northerlies cluster is dominated by northerly winds with warm and moist air throughout its profile (Fig. 1m). Similar to the wet trades, the northerlies display strong ascending motion throughout

the profile. The northerlies are most frequent in January-February, which is the active monsoon period, though only occurring around 20% of the time. From December to February, the monthly distribution of trade and non-trade clusters is consistent with the trade wind clusters present for 56% of each month. By March this ratio shifts to 77% trades and to 92% in April.

When comparing the synoptic environments associated with each cluster, the main differences lie in the location of the

high-pressure systems of the subtropical ridge. In the three trade wind clusters, the high is located over south-east Australia or the Tasman Sea, forming a ridge along the Australian east coast (Fig. 1b, e, h), which triggers the typical southeasterly trade winds. While in the non-trade clusters, in the absence of an eastern high, weak pressure gradients develop over the Australian east coast (Fig. 1k & n).

In the classic trades, the high is the strongest (1020 hPa) of the three trade clusters and centred over south-eastern Australia, while in the summer trades, the high is now weaker (1018 hPa) and centred over the Tasman Sea. In the wet trades, the Tasman high is larger than in the summer trades, creating a more zonally orientated ridge along the east coast, forcing the easterly







**Figure 1.** The left column shows a Skew-T plot of the ERA5 cluster composite soundings for Davies Reef (18.75° S, 147.75° E, blue star). The air temperature is shown in red, while the dew point temperature is shown in blue. The solid lines represent the mean, while the shading shows the upper and lower quartiles. The middle column shows the cluster synoptic surface means. ERA5 data is used for the mean sea level pressure (black contours, 2 hPa spacing) and horizontal winds (quivers, m/s, quiver key found in panel b), while NOAA's DOISST is used for the SSTs (filled contours). The right column shows the 72 hour back trajectory composites launched from the 925 hPa. The composites are normalised and placed on a log scale. The maroon outline marks the boundaries of the Great Barrier Reef.





component to the wet trades' winds. The east coast ridging is strongest in the classic trades and extends furthest north with trade winds covering the entire GBR. The ridging extent and therefore latitudinal range of the trade winds decreases within the

summer and wet trades. The general poleward shift and weaker tropical pressure gradients in the wet trades shows westerlies forming in the Arafura Sea north of Australia, indicating a monsoonal shift. In the doldrums, weak pressure gradients now cover the GBR and northern Australia, while the westerlies in the Arafura Sea strengthen. Finally, the northerlies depict a classic monsoon set-up, with surface westerlies over Darwin brought by the deepening of the low pressure over northern Australia (Troup, 1961). In turn, this low pressure also turns the GBR winds to northerly.


Moving from the surface wind fields, we next considered the origin of these boundary layer air masses. We find the cluster back trajectories largely follow the surface wind fields, where the trade wind clusters have minimal continental influence to the Davies Reef air masses, aside from a small portion in the classic trades that move anticyclonically from the continent to Davies Reef (Fig. 1c, f, i). Air masses in the classic trades also come from the furthest south, with many trajectories extending

into the Southern Ocean. The continental influence is larger in the non-trade clusters. The doldrums have no strong preference in trajectory direction. Most trajectories originate from the neighbouring Coral Sea, with occasional continental influence from far north-eastern Australia (Fig. 1l). The northerlies indicate two main trajectory paths, which are a monsoonal path coming from the northwest via the Maritime Continent and an easterly to north-easterly path from the Coral Sea (Fig. 1o). When considering the pressure levels the air masses originate from (Fig. 3), the classic and summer trades show almost identical

distributions originating from around 800 hPa. However, the wet trades tend to pull air masses from the lower atmosphere (875 hPa) with many trajectories originating near the surface (Fig. 3c). In the non-trades, the doldrums show a stronger subsidence pattern than the wet trades originating around 860 hPa despite the trajectories travelling a shorter horizontal distance. Conversely, the northerlies have the largest surface influence with many trajectories originating from below 950 hPa and on average from 915 hPa (Fig. 3d).


Having now described the basic properties of our clusters, it is important to analyse how the clusters relate through their transition preferences. As each day is assigned to one cluster, we calculate a simple transition matrix (Table 1) that shows the frequency with which each cluster transitions to another on the following day. Our transition matrix is not only useful in

highlighting the common transition paths between the trade winds and non-trade clusters, but also in identifying which cluster transitions seldom occur. Each cluster strongly biases transitioning to itself, with the strongest self-bias in the classic trades at 75%. The three trade wind clusters have a secondary preference for another trade cluster. The classic trades moves to the summer trades, the summer trades moves to the wet trades and the wet trades moves back to the summer trades. To move from the trade wind to non-trade clusters, the most likely path is from the summer trades to doldrums (9%) followed by the wet

trades to northerlies (8%). In the non-trade clusters, the doldrums have a similar preference for transitioning to the wet trades or northerlies. In contrast, the northerlies have the strongest non-self preference to transition to the wet trades (19%). Thus, the wet trades are likely a transition phase from the non-trade to the trade clusters as transitions from the northerlies to the





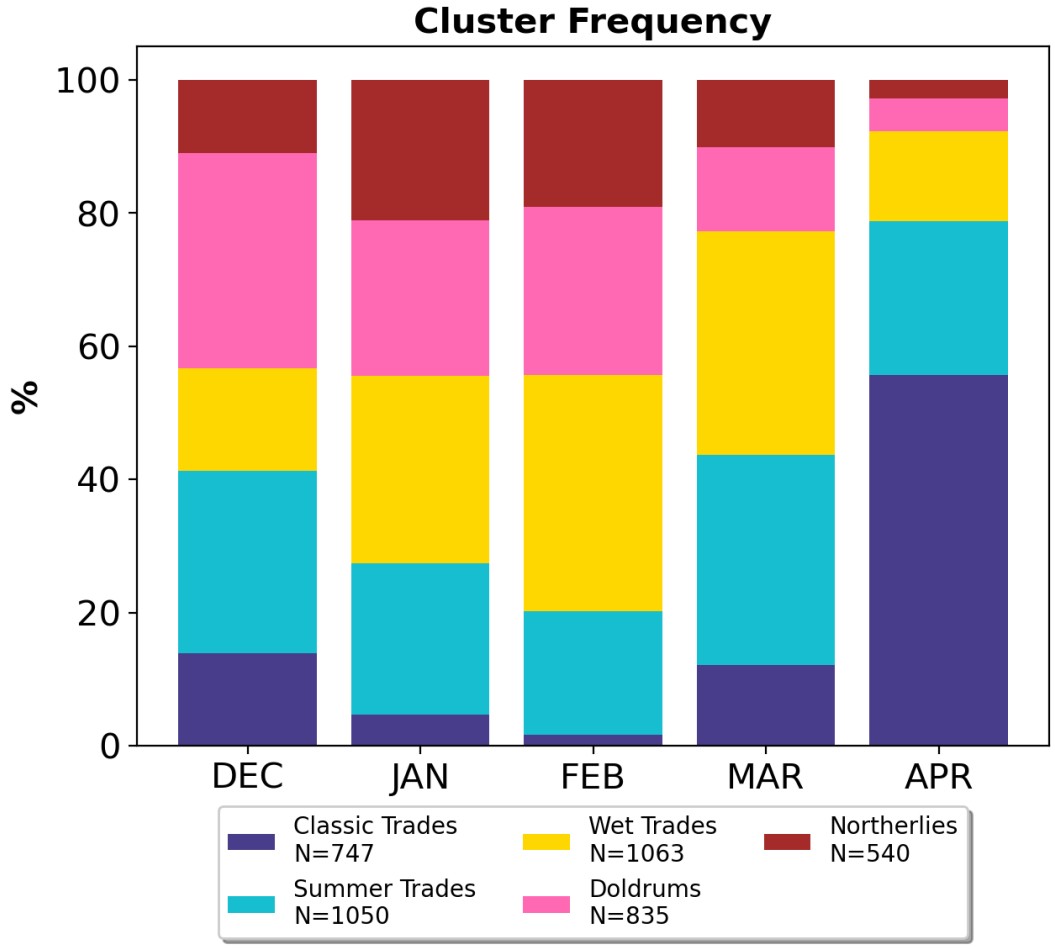

**Figure 2.** Cluster frequency for each month based on the 1996-2024 period.

summer/classic trades are extremely rare (≤1.5%). Overall, it is harder to transition from the trade winds to non-trades (10% average) than vice-versa (22% average).

**3.2 Synoptic anomalies**

To extend our understanding of the typical meteorology during each cluster, we calculate the day-of-year relative anomalies (Fig. 4) for each cluster to remove any seasonal biases. Consistent with typical summer patterns over Australia (Ndarana and Waugh, 2011; O'Brien and Reeder, 2017), many clusters indicate Rossby-wave breaking activity. This is most clearly evident in the doldrums cluster, where a pronounced wave train extends across the Pacific from Australia to South America (Fig. 4g-h).

At the end of the wave train, the cyclonic anomaly over south-eastern Australia at 500 hPa indicates the formation of a deep cut-off low, extending from 250 hPa (not shown) to the surface, as the result of wave breaking on a previous day. While the



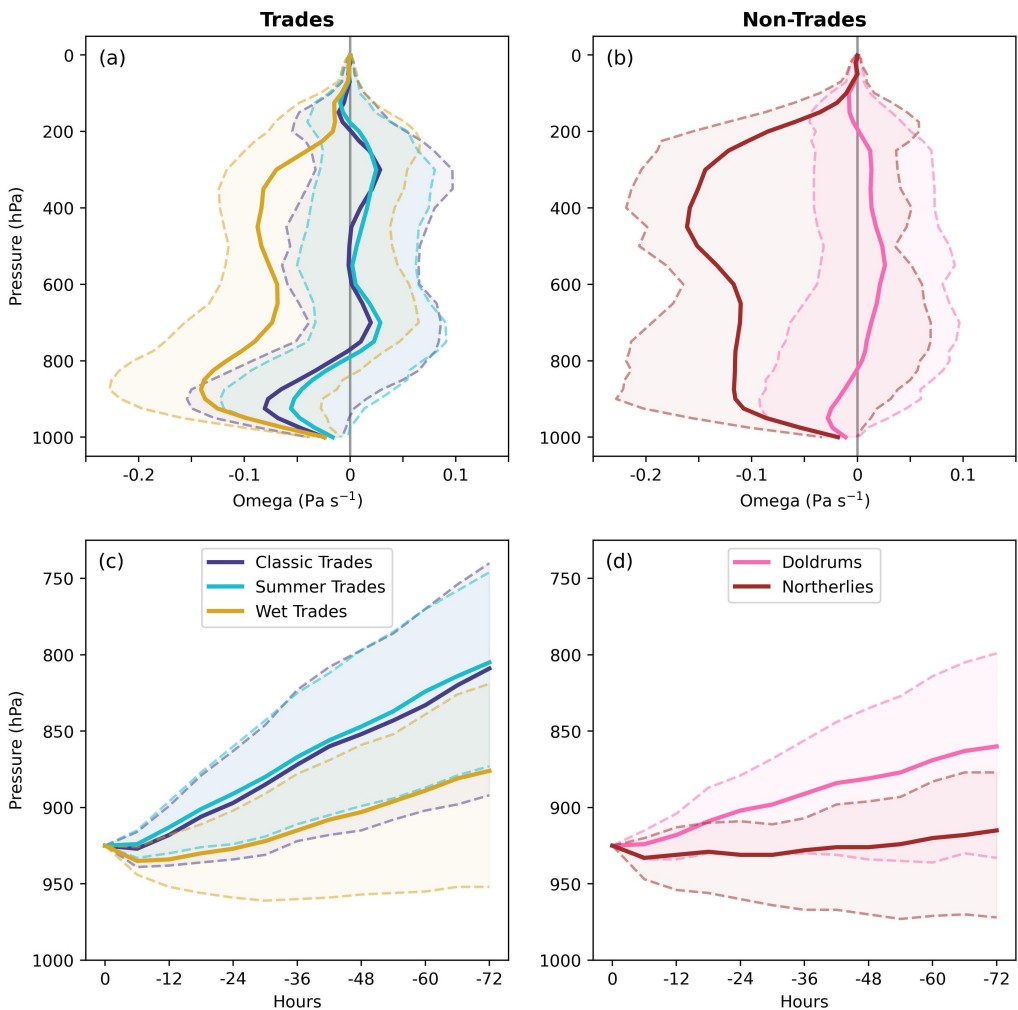

**Figure 3.** The means (solid) and lower-upper quartile ranges (dashed and shaded) are shown for both the ERA5 omega profiles at Davies Reef (a-b) and the pressure changes throughout the back trajectories launched from 925 hPa at Davies Reef (c-d).





|  | | Final cluster | | | |
|---|---|---|---|---|---|
|  | Classic Trades | Summer Trades | Wet Trades | Doldrums | Northerlies |
| Classic Trades | 74.8% | 14.3% | 5.4% | 4.6% | 0.9% |
| Summer Trades | 8.5% | 68.1% | 13.5% | 9.2% | 0.7% |
| Wet Trades | 6.2% | 14.7% | 64.5% | 6.4% | 8.2% |
| Doldrums | 3.1% | 8.5% | 10.8% | 67.3% | 10.3% |
| Northerlies | 1.5% | 0.2% | 19.4% | 13.5% | 65.4% |

*Initial cluster* (row label, left margin)

**Table 1.** Cluster transition matrix showing the transition percentage from the starting cluster (first column) to the final cluster (first row).

northerlies have the same cyclonic anomaly at 500 hPa over south-eastern Australia, the anomaly is weaker and lacks the wave train pattern of the doldrums (Fig. 4i-j). The upper-level cyclonic anomaly sits between the two anticyclonic anomalies creating a dipole structure where the cyclonic anomalies north-west to south-east orientation creates strong northerlies over the GBR
that extends down to the surface.

Over the GBR, the doldrums have the warmest SST anomalies with the highest temperatures radiating out from the central GBR and following the surface cyclonic anomaly to extend into the Coral Sea (Fig. 4g). Here, the cyclonic anomaly is encircled by warmer SSTs over the GBR that extend into the Coral Sea and down to New Zealand, while cooler SSTs are
observed over south-eastern Australia. Conversely, the northerlies cluster shows warm SST anomalies confined mainly to the southern GBR. These anomalies are weaker and centred further east in the Coral Sea, while cooler SSTs expand from the north-western Australian coast to the Maritime Continent (Fig. 4i). This cooling pattern is likely driving by strong westerlies over northern Australia, which intensify wind-evaporation feedback and promote SST cooling (Sekizawa et al., 2018, 2023).

The three trade clusters all show general anticyclonic anomalies at the surface centred over south-eastern Australia. In the classic trades, the anticyclone is stronger and sits over Tasmania, while in the summer and wet trades, the anticyclone sits further north forming an inverted-u-shaped anomaly (Fig. 4a, c, e). The anticyclones in the classic and summer trades extent vertically to at least 500 hPa, indicating deeper systems (Fig. 4b, d). In the classic trades, a cyclonic anomaly tongue extends over the GBR creating the westerly wind reversal seen in the cluster soundings (Fig. 1a). This tongue becomes more defined at
250 hPa (not shown) signalling the presence of wave breaking found at a higher level compared to the doldrums. The classic trades show the strongest cool SST anomalies over the GBR which extends down the east coast and towards New Zealand, while in the summer trades, the cool SSTs sit further east in the Coral Sea extending only to the northern GBR. In contrast, the wet trades display weaker and less coherent upper-level anomalies (Fig. 4f), likely due to enhanced convection in the GBR





**Figure 4.** Day-of-year based cluster anomalies for the surface (left) and 500 hPa (right). The surface anomalies show ERA5 mean sea level pressure (contours, 1 hPa intervals) and 10 m horizontal winds (quivers, m/s) with DOISSTs (filled contours, $^\circ$C), while at 500 hPa, only the ERA5 geopotential height (filled contours, 10 m intervals) and horizontal winds (quivers, m/s) are shown. A statistical significance mask of $p<0.01$ has been applied to the SST anomaly contours.





region. As the wet trades is also considered a transition cluster between the non-trade and trade clusters, this may contribute to the weaker anomaly signals. Additionally, the presence of near-neutral SST anomalies over the GBR in the wet trades suggests this cluster does not strongly bias toward either ocean cooling or warming.

### 3.3 Davies Reef observations

Moving next to the local analysis at Davies Reef, we connect the observations from the Davies Reef AWS (Table 2) and Himawari-8/9 satellite imagery (Fig. 5) to the cluster surface energy budget (Fig. 6) at the same site. The most pronounced difference in the net surface energy budget between the trade wind and non-trade clusters lies in the LHF, with values in the non-trades being roughly half that of the trade wind clusters. Additionally, all trade wind clusters have negative net fluxes indicating ocean cooling, while the non-trade clusters show positive net fluxes (ocean heating).

The trade wind clusters are marked by strong south-easterly winds that produce a large LHF and overall negative net flux (ocean cooling). Among them, the classic and summer trades have similar surface observations, with the summer trades distinguished by warmer ocean and air temperatures. The summer trades also exhibit reduced cloud fraction (44%) compared to the classic trades (59%), primarily due to a decrease in high cloud (mainly cirrus), and consequently an increased SWF. The wet trades differ significantly from the classic and summer trades featuring both stronger surface winds and the second-highest relative humidity and rainfall. The wet trades also shows the highest cloud fraction (77%) with prominent amounts of high cloud (38%), deep convection (6%), and alto-stratus (11%). Despite the stronger surface winds, the higher humidity in the wet trades has weakened the cluster's LHF, however, it is still within a normal range.

The non-trade clusters are set apart by their warm ocean and air temperatures, weaker surface winds blowing from the northeast, and reduced LHF allowing for a positive net flux (ocean heating). The doldrums cluster captures the calm and clear conditions often described in the literature (Klocke et al., 2017), producing the warmest ocean and air temperatures, the weakest winds, and lowest cloud fraction (41%). Reductions in low-cloud coverage, particularly stratocumulus, differentiate the doldrums from the summer trades, where the small increase in deep convection indicates the passage of infrequent storms not captured in the classic or summer trades. Expectantly, the doldrums produce the highest SWF and lowest turbulent fluxes. Despite similar average wind directions, the northerlies are considerably more humid than the doldrums, also boasting the highest rainfall and cloud fraction (81%) despite having the lowest proportion of low clouds (8%). Whilst the cloud fraction is similar to the wet trades, the northerlies produce significantly more high cloud (57%), built by large amounts of deep convection (17%) and cirrostratus (26%). This increased cloud fraction causes a weaker SWF and net flux compared to the doldrums.






**Figure 5.** Decmber to April cluster average cloud type using JAXA Himawari-8/9 cloud typing for a 1° box centred on Davies Reef at 0000 UTC (1000 LST). Cloud typing is based on the ISCCP cloud classification system. The high clouds are represented by the purples, mid-level clouds the oranges, and low clouds the blues/greens. Note data is only available from December 2015 to April 2024.





| | Classic Trades | Summer Trades | Wet Trades | Doldrums | Northerlies |
|---|---|---|---|---|---|
| | 747 days | 1050 days | 1063 days | 835 days | 540 days |
| 4 m Ocean temperature (°C) | 27.2 | 28.1 | 28.3 | 28.7 | 28.6 |
| Air temperature (°C) | 26.1 | 27.2 | 27.3 | 28.2 | 27.6 |
| Wind speed (m/s) | 8.3 | 7.7 | 9 | 4.4 | 6 |
| Wind direction (°) | 107 | 100 | 103 | 27 | 31 |
| Relative humidity (%) | 68.8 (311 days) | 68.3 (539 days) | 76.4 (539 days) | 74 (442 days) | 80 (280 days) |
| Rainfall accumulation (mm/day) | 2.4 (339 days) | 1.5 (598 days) | 12.8 (630 days) | 1.7 (520 days) | 26.5 (333 days) |

**Table 2.** AIMS Davies Reef AWS daily averages. The atmospheric variables are taken at 12 m. Relative humidity data is only available from 2008-2022, and rainfall 2008-2024. The amount of days captured by the reduced data sets is shown for each cluster.

## 3.4 MJO

The MJO is known as a key driver of tropical convection and wind patterns, thus it may be insightful to analyse the MJO phase distribution between the clusters. In this study, the MJO is divided into its eight standard phases along with corresponding weak (inactive) phases (Fig. 7). Over the Australia region, phases 4-7 are considered to be the enhanced convection phases and 8-3 the suppressed convection phases (Wheeler and Hendon, 2004). Although Davies Reef sits outside of the latitude range considered in the MJO definition (15° N-15° S), MJO-related influences remain relevant. Specifically, phases 5-6 are expected to produce

enhanced convection over the northern GBR, and phases 1-2 are generally associated with suppressed convection in this region.

The trade wind clusters have a comparatively even distribution across the eight MJO phases. The classic trades show a small bias for phases 1-3 being the only cluster with more days under the suppressed convective phases for Australia (35.7%) and the highest portion of days considered weak (38.7%). The summer trades display the most balanced distribution among all phases,

although a modest preference remains for the enhanced convective phases (34.7%). In the wet trades, phase 3 is most common (11.5%) with phases 4-5 closely behind (10.5% & 10.2%). Phases 8-1 are also less frequent in the wet trades indicating a preference for an active MJO over the western-central Australian tropics. Overall, 38.6% days fall into the enhanced convective





**Figure 6.** Cluster averages of daily averaged net surface fluxes (bars) with the total net flux (black) and AIMS Davies Reef AWS 4 m ocean temperature (blue). Note here positive (negative) flux values represent incoming (outgoing) radiation.

phases, while only 29.3% are under the suppressed phases.

The non-trade clusters show clear bias for phases 6-7 with both clusters preferring phase 7 which denotes an active MJO in the western Pacific. The northerlies show the largest frequency in phase 7 (16.9%), closely followed by the doldrums (14.4%). While the proportion of days in phases 4-5 is not remarkably different between the non-trades and trade wind clusters, there is clear preference for phases 6-7 and away from the suppressed phases (particularly phase 2). This indicates that the non-trade clusters are more strongly associated with active MJO conditions over the central-eastern Australian tropics, with 41% of days under enhanced convective phases in the doldrums and 47% in the northerlies.



**Figure 7.** MJO phase breakdown for the December-April 1996-2024 period separated into active phases (solid) and weak phases (hashed). Based on the Australian region, the suppressed convective phases 8-3 (browns) and the enhanced convective phases 4-7 (purples/blues) are grouped.





## 4 Coral bleaching

The doldrums cluster captures weather conditions strongly conducive to mass coral bleaching, being weak winds and clear
skies, thus unsurprisingly produces the warmest ocean temperatures on average. Therefore, we now analyse the proportion of
days in each cluster that exceed normal temperature ranges for our site Davies Reef. Using the local bleaching threshold (29.8
°C) for Davies Reef (Berkelmans, 2002) and the maximum +1 standard deviation temperature (29.4 °C) (Richards et al., 2024)
for Davies Reef, we can indicate the coral bleaching potential in each cluster (Table 3).

|  | Classic Trades | Summer Trades | Wet Trades | Doldrums | Northerlies |
|---|---|---|---|---|---|
| Total days | 747 | 1050 | 1063 | 835 | 540 |
| Days > 29.4 °C | 0 | 15 | 37 | 119 | 56 |
| % of cluster | 0% | 1.4% | 3.5% | 14.3% | 10.4% |
| Days > 29.8 °C | 0 | 0 | 7 | 29 | 18 |
| % of cluster | 0% | 0% | 0.7% | 3.5% | 3.3% |

**Table 3.** Number of days the Davies Reef daily averaged 4 m ocean temperatures exceeds the maximum +1 standard deviation temperature
(29.4 °C) and local bleaching threshold (29.8 °C) during the 1996-2024 December-April period for each cluster.

As the doldrums cluster captures the weather associated with coral bleaching and most frequently exceeds the local bleach-
ing threshold at Davies Reef, it would be reasonable to conclude that mass coral bleaching years are related to spikes in
doldrums frequency. After separating the cluster trends into bleaching ('98, '02, '06, '16, '17, '20, '22, '24) (AIMS, 2025)
and non-bleaching years (Fig. 8), bleaching years have on average nine more doldrums days per year (Fig. 8a). By removing
the fringe months and focusing on the warmest months of the bleaching season (January-March) when the doldrums are more
frequent (Fig. 2), the doldrums stand out as the main difference between bleaching and non-bleaching years (Fig. 8b). Interest-
ingly, the northerlies show minimal difference between bleaching and non-bleaching years and are at times more frequent in
non-bleaching years. While still being a non-trade cluster associated with heating, the northerlies, unlike the doldrums, do not
seem to cause a normal year to become a bleaching year, and may simply form in the already warmer months (Fig. 2).

Along with an increase in the doldrums, there is a notable absence of the strongest cooling cluster, the classic trades, dur-
ing mass bleaching years which have instead been replaced with the weaker summer trades cluster (Fig. 8 c-d). While the
classic trades are normally relatively infrequent from December-March, in December, five out of eight bleaching years had
no classic trades form. Although there are years that also had no classic trades form in December that did not bleach (2011
& 2019), these years had strong cooling in April with 26 and 18 classic trades forming respectfully. Comparatively, all eight
bleaching years did not exceed 14 classic trades in April. As the classic trades normally make up 56% of the April cluster,
bleaching years may be experiencing an unusual extension of the warm season in conjunction with an earlier weakening of the
trade winds in December.





**Figure 8.** Cluster frequency box plots splitting the 1996-2024 data into mass bleaching ('98, '02, '06, '16, '17, '20, '22, '24) and normal years. The distribution is split into the full GBR coral bleaching season (December-April) (a), isolating the warmest months (January-March) (b) and the fringe months; December (c) and April (d).





While we do see evidence of more doldrums and fewer classic trades in bleaching years, it is important to note that given the number of recorded CBE cases is relatively limited (eight CBEs between 1996-2024), this does constrain the statistical robustness we can produce from our comparison between bleaching and non-bleaching years.

## 5  Case Studies

While on average the non-trade clusters, in particular the doldrums, show the highest risk for coral bleaching, it is important to understand the cluster evolution during CBEs. Here we analyse the clusters across three separate GBR mass bleaching events during the three El Niño Southern Oscillation (ENSO) phases; 2016 (El Niño), 2022 (La Niña), and 2024 (Neutral), where the phase distinction is based on the December-February average Southern Oscillation Index. We analyse daily averages of the ERA5 surface net flux and its four components with 4 m ocean temperature and surface wind observations from the AIMS
Davies Reef AWS (Fig. 9). As it is difficult to determine the exact start/end dates for individual CBEs, we instead analyse a 91-day period centred on the warmest day in the 4 m ocean temperature record for each CBE.

In general, all three bleaching events show the same well-documented meteorological pattern where periods of ocean temperature spikes coincide with a change in wind direction away from the south-easterly trades and a drop in the LHF forced
by weakened surface winds and/or increased surface humidity, regardless of the phase of ENSO. These heating periods are followed by relatively rapid cooling driven by a return to the stronger trade south-easterlies and subsequent enhancement of the LHF. However, the timing and duration of these heating periods differ across the events.

In 2016, there were three distinct temperature spikes with gradual warming during periods of calm winds and rapid cool-
ing as strong easterly winds re-established on February 8th, February 21st, and March 2nd (Fig. 9a-b). These events clearly illustrate the relationship between wind speed and LHF. Each warming spike was associated with dominance of non-trade clusters (primarily the doldrums), followed by a shift to wet or summer trade clusters during the subsequent rapid cooling phase. The first heating spike in particular saw a large consecutive non-trade period lasting 15 days (6 doldrums, 9 northerlies), where the ocean temperature rose 0.9 °C. Conversely, the 2022 CBE featured a single major heating spike that hit a maximum
on March 10th (Fig. 9c-d). This heating period also coincided with a stretch of 15 consecutive non-trade wind clusters (12 doldrums, 3 northerlies) resulting in a 1.2 °C temperature increase before transitioning into the wet trades and then the classic trades as temperatures plummeted. Unlike 2016 and 2022, the 2024 heating appeared earlier in the season with a first spike in late December and a second spike in late January (Fig. 9e-f). The December spike was particularly intense, corresponding to the longest consecutive non-trade period (18 days, 13 doldrums and 5 northerlies), during which ocean temperatures rose by 2.4 °C.


The cluster distribution across the three periods shows that the doldrums are the most frequent (32%), while the wet trades fol-





**Figure 9.** Case studies during the 2016, 2022 and 2024 GBR CBEs taken at Davies Reef on a daily average time frame. Panels (a), (c), (e) show ERA5 flux data, with net short-wave (pink), net long-wave (green), latent heat (blue), sensible heat (brown) and the net flux (black). Here positive (negative) values represent incoming (outgoing) radiation. Panels (b), (d), (f) show surface wind speeds (grey, m/s) and directions (barbs, m/s) and 4 m ocean temperatures (blue, °C) from the Davies Reef AIMS AWS. The daily cluster is plotted along the temperature.





low closely (30%) with the summer trades (17%), northerlies (16%), and classic trades (5%). The absence of the classic trades is unsurprising as they are typically missing until April (Fig. 2). The doldrums are not only the most frequent but generally persist the longest with an average duration of 3.9 days with 30% of events ≥6 days. Conversely, the wet trades have an average duration of 2.6 days with only 13% ≥6 days. Similarly to the results in cluster climatology, the non-trade clusters show positive temperature gradients, with the strongest average heating (0.09 °C/day) and warmest average ocean temperatures found in the doldrums (29.2 °C), while cooling mainly occurs in the trade wind clusters, particularly the wet trades (-0.08 °C/day).

## 6 Discussion

While the ocean temperature evolution across the three mass CBEs case studies differs greatly, the cluster distribution during heating and cooling events paints the same picture. Periods of ocean heating align closely with drops in the LHF brought about by decreased wind speeds and wind shifts away from trade wind conditions (Fig. 9), which is well documented in studies of both the 2016 (Bainbridge, 2017; Benthuysen et al., 2018; Karnauskas, 2020) and 2022 (McGowan and Theobald, 2023; Richards et al., 2024) CBEs. The evolution of clusters during these ocean heating periods also shows a preference for sustained non-trade conditions, particularly long stretches of doldrums periods, where the largest temperature spikes in each CBE saw non-trade clusters persist between 15-18 days. Conversely, periods of ocean cooling tend to occur once the trade winds re-establish, bringing stronger winds and more cloud cover over Davies Reef, facilitating ocean heat loss, causing ocean temperatures to drop. Notably, the most pronounced temperature drops tend to occur during the wet trades cluster, likely due to the fact that both doldrums and northerlies most frequently transition into this cluster (Table 1). When comparing the eight CBEs, bleaching years not only had more doldrums days during the warmer January-March period but also less of the classic trades in December and April which generally bring ocean cooling. A lack of stronger trades in December could allow ocean heat to build up early in the season (Spady et al., 2022), which would be amplified in the warmer January-March period as the trades break down more frequently into the doldrums. Likewise, the lack of classic trades in April may allow the warmer ocean temperatures to persist for longer causing late-season bleaching (Smith and Trewin, 2024).

While it is clear non-trade conditions provide the ideal environment for ocean heating, what causes the initial breakdown and duration of the non-trade clusters is more complex. Given that trade winds represent the climatological baseline for the region's meteorology (Malkus, 1958), their breakdown requires the introduction of a synoptic-scale disturbance capable of disrupting the subtropical ridge pattern that sustains them. For the doldrums, this disturbance needs to flatten the pressure gradients over the GBR region. One potential mechanism is Rossby-wave breaking over eastern Australia, where the associated potential vorticity mixing can weaken surface pressure gradients and suppress trade wind flow (Hoskins et al., 1985). Our synoptic anomalies do indicate Rossby-wave activity through the presence of a wave train that has just passed over eastern Australia (Fig. 4h). Consecutive anticyclonic Rossby-wave breaking events were noted during the 2022 bleaching event and thought to drive the breakdown of the trade winds in this event (Richards et al., 2024) while Rossby wave have also been connected



to marine heatwaves in the North Pacific (Di Lorenzo and Mantua, 2016) and western Indian Ocean (Chambers et al., 1999; Spencer et al., 2000). Interestingly, our wave chain is almost identical to the work of Lee et al. (2010), who attributed a central South Pacific marine heatwave in 2009-2010 to the overlying anticyclones in a Rossby-wave chain. This process may also be similar to the doldrums in the North Atlantic trade region where the location of Rossby-wave breaking has also been found to
alter the typical trade wind flow during boreal winter when the trades are usually more stable (Aemisegger et al., 2021). The passage of African Easterly Waves is also thought to weaken the trade winds in the tropical east Atlantic, allowing the doldrums to form creating local SST heating (Thiam et al., 2025). While these interactions with Rossby-waves provide valuable insight into the synoptic patterns over the GBR, further research is needed to fully understand their influence on the trade wind regions.

In contrast to doldrums, the transition to the northerlies requires northerly winds over our GBR site, typically driven by a low-pressure area forming south of Davies Reef. Our synoptic anomalies at 500 hPa show a dipole structure creating a region of enhanced northerly winds over the GBR, while the origin of the structure remains uncertain, it may be linked to tropical convection in the region. Both non-trade clusters showed a preference for the MJO in phases 6-7 which was not present in the trade wind clusters. Perhaps the passage of the MJO over the northern GBR could contribute to the breakdown of the trade
winds on some occasions, however, it does not appear to be a sole driver, as while we do see enhanced convection and rainfall in the northerlies, we find the opposite in the doldrums. Here, the existing local circulation at Davies Reef, particularly the lower tropospheric winds, appears to determine how the local environment interacts with the MJO as only the northerlies have the existing northerly winds that are capable of transporting the moist air from the tropics. This is also evident in the back trajectories, as the northerlies contain air masses from either the Maritime continent or neighbouring Coral Sea that are found closer to the
surface, while in the doldrums air masses rarely originate equatorwards of 15° S (Fig. 1l) and are generally subsiding (Fig. 3), as was proposed by (Windmiller, 2024). While it has been argued that a faster propagating MJO in phases 6-7 can disrupt these doldrums periods (Gregory et al., 2024), that is not obvious in our cluster analysis, where the doldrums also favour phases 6-7. As the MJO is also only defined for the tropics (15° N-15° S) we cannot expect the same phase relationship with sites outside of this region (Wheeler et al., 2009). For example, Dao et al. (2025) showed how the local environment can enhance precipi-
tation over the Townsville region (100 km southwest of Davies Reef) even during the suppressed convection phases of the MJO.

The transitions between the five clusters and the trade wind and non-trades regimes is a complicated process, however, we propose a basic 'trade wind cycle' to describe the general transitions between the local meteorology (Fig. 10). Trade winds represent the basic state of the regional atmosphere, making them more resistant to disruption and easier to re-establish once
perturbed. Our transitions suggest that wind regime clusters generally transition between the summer and wet trades until the system is disturbed. Then it shifts into the non-trade regime, typically from the summer trades to the doldrums where the amount of low cloud drops as the winds weaken while the temperature increases. The system then move to the northerlies where the low cloud further dissipates due to the high humidity and low LHF while the high cloud and deep convection dominate the region. Re-establishment of trade winds generally occurs through the wet trades cluster. During this phase, the return
of easterly winds leads to a drop in humidity and an increase in LHF, fostering shallow convection. However, high cloud and



deep convection from the preceding northerly phase often remain. Although the transitions between wind regimes are inherently complex, the integration of the transition matrix with the proposed 'trade wind cycle' provides meaningful insight into the processes underlying ocean heat and atmosphere variability. This framework offers valuable potential for improving the prediction of particular wind regime, as well as informing long-term strategies for mitigation of CBEs.


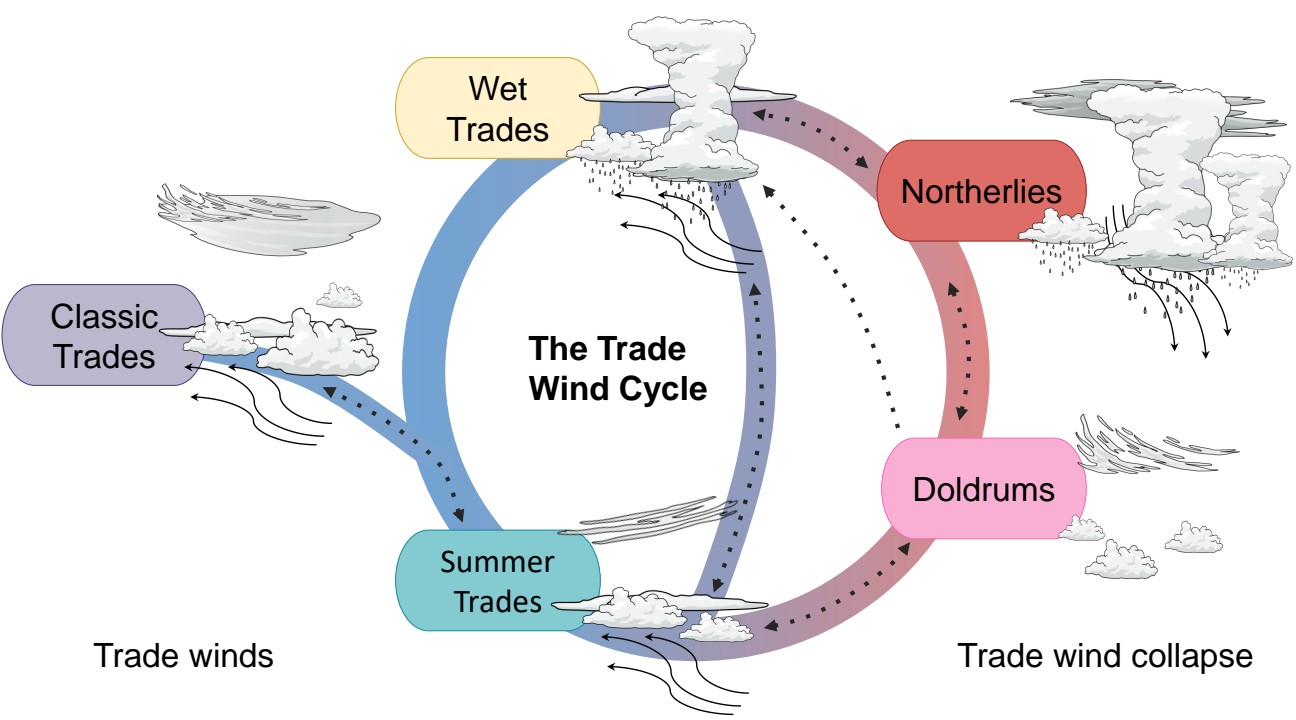

**Figure 10.** Schematic of the proposed trade wind cycle.

As the shallow waters of coral reefs are particularly sensitive to changes in the daily meteorology, the ability to forecast these events is again dependent on how well these weather-scale processes are represented in subseasonal prediction models. Over the GBR, while there is skill in forecasting the start of the marine heatwave, there is less predictive skill in determining the marine heatwave's end due to the under representation of weather scale processes like tropical cyclones (Benthuysen et al., 2021). While our study has begun to explore the relationship between the trade winds and ocean temperature, there is a clear need for better process-level understanding of ocean-atmosphere interactions to improve the prediction of bleaching. Particularly, improving the predictive skill of the events' end or the duration of the thermal stress period would provide essential information for reef management and regional climate adaptation strategies. In turn, influencing the continued survival and



health of the GBR and the countless stakeholders dependent on it.

## 7   Conclusions

Through our clustering climatology of the GBR coral bleaching season, our study has shown how the dominant synoptic patterns can be separated into a trade wind and non-trade wind regimes. The three identified trade wind clusters, which typically

bring stronger winds and more extensive cloud cover, are associated with ocean cooling while the non-trades, particularly the doldrums, have weak LHFs brought by weaker winds and higher humidity, that cause ocean heating.

The doldrums capture not only the warmest GBR ocean temperatures but have the highest risk of coral bleaching with 3.5% of days exceeding the local bleaching threshold at Davies Reef. The frequency of the doldrums is also important for differ-

entiating a mass bleaching year from a non-bleaching year. During January-March when ocean temperatures are warmest, years with mass bleaching have on average nine more doldrums days than non-bleaching years. In conjunction, mass bleaching years more often have no classic trades in December combined with low numbers in April meaning less ocean cooling during these months. This could contribute to a build-up of early summer heat stress and/or the persistence of heat later into the season. Thus, while it was already apparent that periods of calm and clear conditions which are brought by the doldrums cause

ocean heating, the absence of cooling events on the edges of the coral bleaching combines to elevate the risk of mass bleaching.

To transition into the doldrums the trade winds over the GBR need to be disrupted. The most prominent mechanism in our analysis is Rossby-wave breaking which can weaken the pressure gradients forming areas of weak winds, clear skies, and subsiding air. However, there are other mechanisms likely to break down the trade winds such as the passage of tropical cyclones

or jumps in the monsoon trough which should be investigated further.

This work provides a framework for understanding the evolution of the trade winds over the GBR, whose presence is important for ocean cooling and preventing marine heatwaves. As the local weather conditions are essential drivers of GBR ocean temperatures, furthering our understanding of the trade wind structure and how they break down is crucial for the continued

monitoring and forecasting of marine heatwaves and thermal bleaching events.



*Data availability.* All datasets used in this study are publicly available online. The Australian Institute of Marine Science Davies Reef data is available at https://doi.org/10.25845/5c09bf93f315d (Australian Institute of Marine Science (AIMS), 2020). The ERA5 reanalysis data is available from the Copernicus Climate Change Service (C3S) Climate Data Store (CDS), with single level data found at

https://doi.org/10.24381/cds.adbb2d47 (Hersbach et al., 2023a) and pressure level data at https://doi.org/10.24381/cds.bd0915c6 (Hersbach et al., 2023b). The Himawari-8/9 cloud categorisation data is available from the Japan Aerospace Exploration Agency (JAXA) P-Tree System (https://www.eorc.jaxa.jp/ptree/). The Bureau of Meteorology SOI data is available at http://www.bom.gov.au/climate/enso/soi/ and the MJO RMM index from http://www.bom.gov.au/climate/mjo/. Lastly, NOAA's OISST data can be found at https://www.ncei.noaa.gov/products/optimum-interpolation-sst (Huang et al., 2021).

*Author contributions.* LR performed the analysis and prepared the draft manuscript. SS, YH, WZ, and DH supervised and reviewed the manuscript.

*Competing interests.* The authors declare no competing interests.

*Acknowledgements.* This work was supported by the Reef Restoration and Adaptation Program. The Reef Restoration and Adaptation Program is funded by the partnership between the Australian Governments Reef Trust and the Great Barrier Reef Foundation. The authors would

like to acknowledge the Traditional Owners of the Great Barrier Reef, particularly the Wulgurukaba and Bindal people of the Townsville region near the area of our case study. This research is also supported by the ARC Centre of Excellence for Climate Extremes (grant no. CE170100023) and the ARC Centre of Excellence for 21st Century Weather (grant no. CE230100012). We acknowledge the National Computational Infrastructure for their provision of computational resources and services. Yi Huang and Steve Siems are further supported by an Australian Research Council Discovery Grant (grant no. DP230100639). We are grateful for the help of Chenhui Jin for providing assistance

in running and producing the LAGRANTO trajectories. We also acknowledge the help of Michael Barnes for providing useful discussion and interpretation of the synoptic structures, which greatly improved the manuscript.



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
