# Peer review of "Trade wind regimes during the Great Barrier Reef coral bleaching season"

_EGUsphere, 2025_

## Author Comment (AC1)

We would like to thank both reviewers for their constructive and thoughtful feedback. The comments provided have greatly improved our manuscript. Please see the attached file for our responses to all comments.

**Response to Alexander Sen Gupta**

 Except for one analysis related to SSTSA composites, there is no attempt to look at the statistical robustness of results. For example, differences in properties between clusters should be tested.

We have implemented more statistical analysis of our results as suggested. For the composite anomalies shown in Figure 1 (SSTA) and Figure 4 (dSSTA/dt, horizontal winds, MSLP and geopotential height) a simple t-test was used to show where the composite anomalies are statistically significant from zero at either the 95th% or 99th% confidence level.

We have also applied statistical testing to Figure 8 where we use Kolmogorov–Smirnov testing to show whether the cluster frequency differences between bleaching and non-bleaching years are statistically significant to at least p

New appendix Figure S1: Box and whisker plots of the Davies Reef variables (excluding wind direction) to show the distributions within clusters. Dashed lines represent the means while solid lines show the medians.

New Figure 1 (SSTAs have replaced the mean SSTs)

New figure 4 Day-of-year based cluster anomalies for the surface (left) and 500 hPa (right). The surface anomalies show ERA5 MSLP (contours, 1 hPa intervals) and 10 m horizontal winds (quivers, m/s) with the SSTA tendency (filled contours, \$^\circ\$C), while at 500 hPa, only the ERA5 geopotential height (filled contours, 100 gpm intervals) and horizontal winds (quivers, m/s) are shown. Masks have been applied to both the SSTA tendency and horizontal winds where only mean anomalies that are statistically significantly different from 0 at the 95th\% confidence level are shown. For the MSLP and geopotential height contours,

- hatching represents where the composite anomalies are statistically significantly different from 0 at the 99th\% confidence level.
- In figure 4 you show the SSTA associated with different clusters. However your clusters are most closely linked to wind strength conditions and therefore LHF anomalies and warming rate. I suspect that you would get a stronger and more informative plot if you looked at a composites of SSTA tendency (dSSTA/dt).

The SSTA previously shown in Fig.4 has been changed to the SSTA tendency with a p

Fig S1 Skew-T plots of the ERA5 cluster composite soundings for Heron Island (southern GBR) with N representing the number of days in each cluster. Here the air temperature (red) and dew point temperature (blue) are shown with solid lines marking the mean and shading the upper and lower quartiles.

**Lizard Island clustering**

Fig S2 Skew-T plots of the ERA5 cluster composite soundings for Lizard Island (Northern GBR) with N representing the number of days in each cluster. Here the air temperature (red) and dew point temperature (blue) are shown with solid lines marking the mean and shading the upper and lower quartiles.

L80: Day-of-year based anomalies are calculated using a daily mean climatology for the 1996-2024 period to remove the cluster's seasonal biases

Not clear what you mean. The clusters themselves are calculated based on absolute variables rather than anomalies. Seems like you are just saying that you remove the seasonal cycle

Yes, we remove the underlying seasonal cycle. We have clarified this in the revised manuscript.

L83: and have then undergone a simple one sided t-test where only the anomalies with a p-value < 0.01 are considered

Im also unsure what you mean by this. You might use a t-test to see if the composite of a set of anomalies is different from 0, but Im not sure what you are referring to here

Line updated to 'All anomalies have then undergone a simple one sided t-test to show that the mean anomalies are significantly different from zero' for clarity

L85: The daily back trajectories each start at 0000 UTC running for 72 hours from Davies Reef at 925 hPa

It might help to provide some motivation as to what you are trying to achieve with this analysis.

In lines 89 and 167 of the revised manuscript we state our back trajectories are to 'understand the origin of boundary layer air masses'. Here we are interested if the air mass is continental or oceanic and if it comes from lower or higher in the atmosphere.

L89: Davies Reef AWS

It would be useful to have a brief description of the site e.g. is this inside a lagoon vs open ocean

Line 63 updated to 'we chose Davies Reef (18.83S, 147.63E, mid-shelf lagoonal reef) as the clustering location, as it is both in the central GBR, with more frequent CBEs recorded, and has a highly consistent observational record of both meteorological and ocean parameters through the Australian Institute of Marine Science (AIMS) Davies Reef Automatic Weather Station (AWS) placed within the lagoon (Bainbridge, 2017).'

L96: We note ERA5 was previously found to have a high correlation with the AIMS Davies Reef observations

But ERA5 is an atmospheric reanalysis. Are you now referring to atmospheric variables?

Adjusted to 'We note ERA5 was previously found to have a high correlation with the AIMS Davies Reef atmospheric observations'

L95: total cluster days assigned

Not sure what is meant by this phrase

Changed to 'where the reduced cluster sizes are stated in Table 3'

**L100: daily averaged mean**

**Don't need averaged and mean**

Adjusted to 'Due to the lack of in-situ fluxes, a local net surface energy budget is produced at Davies Reef using a daily average of ERA5's mean net surface flux products'

L110: Here we consider only a 1° box centred on Davies Reef taking only the 0000 UTC observations.

Cloud cover can vary rapidly over timescales much less than a day. Why only take a single time stamp if other times are available?

We took the single timestamp to keep consistency with both the soundings and back trajectory analysis, therefore we can both look at the air mass origins for the clouds at this time frame and see the corresponding typical vertical profile. Additionally, we have looked at a later time step (06Z or 4pm LST) where no substantial differences are noted.

We agree that the diurnal cycles of clouds over the reef is important to understand, however diurnal cycles are currently outside of the current scope of this manuscript.

**L121: be the optimal number**

There are objective measures to find optimal cluster numbers. Are you using these or are you just picking manually? I have no problem with the latter, but if you call it optimal, then you should indicate what you are optimising to make it optimal

Line change to 'In the cluster analysis at Davies Reef, we found five clusters were able to best represent the diversity in the atmospheric profiles.'

Fig 1. Hard to see any differences in SST. Might be more helpful to show SSTA. Indeed I would expect low wind conditions to be most strongly related to temperature tendancy

The mean SSTs have been replaced with the SSTAs from figure 4.

Fig 2 given 18 years of data you could assess whether the changes in cluster proportions are actually significantly different from month to month

We have tested some of the month to month differences as suggested where for example the shift between March to April in the frequency of the Classic Trades is statistically significant (p<0.01) using Kolmogorov–Smirnov testing. However we do not see the importance of presenting this testing in our analysis as we do not reference any trends in the cluster distributions or shifts monthly, but rather to describe the monthly distributions of the clusters.

Additionally, while it is simple to test the more prominent months like the difference between March and April for the Classic trades, testing for significance becomes more complicated when the monthly frequencies are heavily skewed towards zero.

L125 do you mean: are ordered based on the direction and the strength of the surface winds?

Corrected

L126: The classic trades are scarce

Better to be precise, the frequency of classic trade wind days is small ...

Revised as suggested

L127: compares

I think you mean, 'is similar to'

Changed to suggested

L129: wind shift

Do you mean wind reversal?

REvised as suggested

L130: Are you referring to Fig 3?

Yes, additional figure references have been added to this section

L130: slightly weaker surface ascent separating the summer trades ON AVERAGE. In general its important to remind the reader that you are talking about composite means not al members of the composite.

Line 134 altered to 'on average ...'

L131: summer trades are evenly distributed

Here and elsewhere the terminology should be tightened e.g. the monthly proportion of summer trade days remain similar across all months

Changed to suggested

L132: and a larger easterly component

Hard to make this out from the figure. Low level winds look pretty similar across the three clusters

We have also added 'and a larger easterly component to the lower-level winds becoming calm aloft' to this line. The calm winds aloft are what is mainly different to the general clockwise turning with height in the other trade clusters.

L134: The wet trades omega profile follows a similar shape

The wet cluster looks quite different to the others to me. It is associated with ascent throughout most of the column, while the others are associated with decent above 800hPa

Changed to 'The wet trades omega profile shows on average net ascent throughout the column with the strongest average surface ascent of the five clusters'

L136: strong boundary layer temperature inversion

In panel j the temperature decreases monotonically with height. I don't see any temperature inversion, just a weakening of the lapse rate

Sentence has now been revised to '..and a weakening of the boundary layer temperature lapse rate'

137/138. Make sure you always reference your figures

Applied as above, figures 2 and 3 particularly are referenced more

141 the monthly PROPORTION of trade and non-trade cluster DAYS REMAINS ALMOST CONSTANT ...

Changes applied

143: this ratio > this proportion

Changes applied

164: with many trajectories extending into the Southern Ocean

The southern ocean is usually defined as south of 60S. From your density plot it looks like very few trajectories would come from there

Revised to 'Tasman Sea'

Fig 3. If negative omega is associated with ascent, its very surprising that clusters associated with the strongest ascent, the Trades and Northerlies are associated with the weakest upward motion in terms of the lagrangian particles (given that the highest proportion of particles is close to the release location). Could there be some issue with the sign convention here?

Our lagrangian back-trajectories also indicate stronger ascent in the northerles and wet trades. As these are backwards trajectories, parcels originating from below the starting pressure levels correspond to ascending.

197: the same > a similar

Change applied

206: shows warm SST anomalies confined mainly to the southern GBR

Looks like the anomalies primarily sit to the east of the GBR

Changed to 'In the northerlies, the SSTAs are weaker and centred further east in the Coral Sea (Fig. 1n) lining up with the only area of significant heating (Fig. 4i). While only weak positive SSTAs impact southern GBR...'

235: relative humidity and rainfall

Here and elsewhere refer to tables and figures

applied

236: high cloud (38%), deep convection (6%), and alto-stratus (11%)

Is there a particular reason to break up the cloud cover? Why not simply present cloud fraction, this is a far more important metric for you discussion?

The total cloud fraction is important and we do also discuss our cloud cover from this view. However, including the cloud typings is also important as there are few studies that actually break down the cloud fraction into individual cloud types which is essential for our fundamental understanding of the environment.

For example different clouds have different radiative effects, so understanding the cloud typing within a cluster adds further details to our net surface energy budgets.

We can also better describe the boundary layer with cloud typing, e.g. the wet trades and northerlies have stronger surface ascent and rainfall which we can connect to the formation of more deep convection not seen in the other clusters.

Except for the SSTA composite analysis no statistical significance was presented. This is important for looking at whether differences in heat fluxes, MJO phases, cluster frequencies etc are significantly different between different clusters. Otherwise with the presentation of just cluster means its impossible to tell what results are robust.

265: The non-trade clusters show clear bias

Its true that the means are bigger, but are these differences statistically significant?

In this section we have removed language such as 'bias' that suggests statistical significance or imply a trend. Here we are mainly trying to describe where the MJO typically resides in relation to our five clusters.

**Section 4. Coral Bleaching**

This is looking at bleaching risk not bleaching. Indeed I don't think its even really looking at bleaching risk its looking at risk of temperature extremes

L275: Using the local bleaching threshold (29.8°C) for Davies Reef (Berkelmans, 2002) and the maximum +1 standard deviation temperature (29.4°C).

Bleaching thresholds typically relate to cumulative metrics like DHW. The absolute temperature thresholds are just the point when stress starts to accumulate. This is why bleaching often occurs after the periods of maximum seasonal temperatures.

We have adjusted the line to 'Using the local 'bleaching threshold' (29.8 C) for Davies Reef (Berkelmans, 2002), marking the point where heat stress begins to accumulate..'

The terminology is consistent with Berkelmans (2002), and the same threshold definition is employed by AIMS. We emphasise that this section does not attempt to determine whether bleaching occurred, but rather to illustrate the accumulation of thermal stress associated with each cluster.

L280: it may be reasonable to suggest, bit it wouldn't be reasonable to conclude

Change applied

L282: cluster trends

Not sure what you mean by trends.

Changed to days

285: the doldrums stand out as the main difference

What does this mean? Are you saying that doldrums have the largest difference in no.of days compared to the other clusters?

Is this difference statistically significant?

It looks like the frequency (which I assume means number of days) goes from about 18 to 21. This seems like a very small difference to explain whether we are getting mass bleaching or not.

We have applied a Kolmogorov–Smirnov test to assess statistical significance between the bleaching and non-bleaching distributions and for the Jan-Mar period the doldrums differences are statistically significant and have been noted in the revised manuscript.

Differences between classic trades and summer trades bleaching vs non-bleaching years do show statistical significance in dec-apr and apr only. These have been highlighted in the manuscript.

Here is a table to summarise the p-values from all the bleaching vs non-bleaching tests performed.

| p-values | classic trades | summer trades | wet trades | doldrums | northerlies |
|----------|----------------|---------------|------------|----------|-------------|
| Dec-Apr  | 0.0004         | 0.0615        | 0.7046     | 0.1588   | 0.9562      |
| Jan-Mar  | 0.8164         | 0.9999        | 0.5084     | 0.0441   | 0.7046      |
| Dec      | 0.0615         | 0.0441        | 0.4191     | 0.9562   | 0.9562      |
| Jan      | 0.5084         | 0.9979        | 0.8884     | 0.1156   | 0.8164      |
| Feb      | 0.8884         | 0.9837        | 0.8164     | 0.2661   | 0.5084      |
| Mar      | 0.8164         | 0.7046        | 0.9837     | 0.8164   | 0.8884      |
| Apr      | 0.0004         | 0.0037        | 0.6074     | 0.8164   | 0.9562      |

L302: this does constrain the statistical robustness

With 8 bleaching and 10 non bleaching you can still conduct a statistical analysis.

Applied as above

L305: show the highest risk for coral bleaching

As above I don't think this is sufficiently precise. Moreover I don't really agree that you are looking a bleaching risk, you are looking at risk of temperature spikes. Bleaching risk would relate to accumulated stress.

We have removed references to 'bleaching risk' and have changed the wording to 'thermal spikes'.

307: 2016 (El Niño), 2022 (La Niña), and 2024 (Neutral)

Did bleaching actually occur on this reef in those years?

The aim here is not specifically to correlate the temperature spikes with local bleaching at Davies Reef, but to understand the cluster evolution throughout a known bleaching period on the central GBR (e.g. what clusters occur on the heating and cooling spikes).

But more specifically, Davies Reef did bleach in 2022, but was not surveyed in 2016. Surveys of other mid-shelf reefs in the Townsville region show low levels of bleaching with the neighbouring Lynch and Centerped reefs reporting 5-10% bleaching that year. (<a href="https://www.aims.gov.au/reef-monitoring/townsville-sector-2016">https://www.aims.gov.au/reef-monitoring/townsville-sector-2016</a>). Bleaching was not reported at Davies Reef in 2024, but was reported at neighbouring reefs (e.g. Myrmidon reef, chicken reef, helix reef).

315: In general, all three bleaching events show the same well-documented meteorological pattern where periods of ocean temperature spikes coincide with a change in wind direction

It is the increase in temperature leading up to a peak, not the peak itself, that should be associated with positive heat fluxes (and typically weak winds). This is why the net heat flux is strongly anti correlated with the wind speed in you figure (not with temperature)

We agree with what you have stated and believe this is currently conveyed in our manuscript as we are analysing the heating and cooling periods not the peaks themselves. Here temperature spikes refers to the whole heating period, not just the peak.

324: The first heating spike

The first heating spike occurs right at the start of the timeseries. Id suggest providing a date

Line adjusted to '..with a first spike on December 19th and a second spike on January 27th..'

334: The doldrums are not only the most frequent but generally persist the longest with an average duration of 3.9 days with 30% of events ≥6 days

Are you referring to the 3 years or in general across the full timeseries?

This question applies for the whole paragraph, please make it explicit

Across the three time series, this has been made more explicit in the paragraph.

335: Similarly to the results in cluster climatology

Not sure what you mean, it would help to refer to figures.

This section was removed from the manuscript

337: temperature gradients > temperature trends

L337: (0.09°C/day) (-0.08°C/day) (29.2°C)

Needs more explanation.

This section was removed from the manuscript

L349: the most pronounced temperature drops tend to occur during the wet trades cluster, likely due to the fact that both doldrums and northerlies most frequently transition into this cluster

I don't understand the logic of this statement

We mean to say you often transition from the non-trades (which are warmer) to the trades (which are cooler) through the wet trades. Therefore this transition period in the wet trades would be when the cooling rate is largest

L367: Lee et al. (2010), who attributed a central South Pacific marine heatwave in 2009-2010 to the overlying anticyclones in a Rossby-wave chain

This RW train in Lee's study was associated with a strong central Pacific warming. Indeed you see a hint of tropical warming in Fig 4g. I wonder if this is the RW source. Would be interesting to extend the domain of that figure a little further to the north.

Interesting yet beyond the scope of the current paper, but we are working on this in the next paper that focuses more on doldrum durations and the origins of this wave. We have made some adjustments to this section. A recent study by Barnes et al. (2025) shows a similar wave propagation from cyclonic Rossby-wave breaking. This is also being investigated more in the next paper.

Barnes, M. A., M. J. Reeder, and T. Ndarana, 2025: Rossby Wave Breaking Morphologies on the Southern Hemisphere Dynamical Tropopause. J. Climate, 38, 4825–4844, https://doi.org/10.1175/JCLI-D-24-0461.1.

We have added 'While the wave in Lee et al. (2023) was associated with strong central Pacific warming, this wave could also be generated by cyclonic Rossby Wave breaking as described by Barnes et al. (2025).' to line 364.

Figure 10: might be worth indicating the proportion of time spent in each of the clusters (e.g. by scaling the cluster box)

We have tested this suggestion, but find it does not make a notable difference to the original figure as many of the proportions in the clusters are similar aside from the northerlies.

397: Why would low cloud dissipates due to the high humidity and low LHF?

The high humidity helps weaken the LHF which helps reduce turbulent convection.

423: with 3.5% of days exceeding the local bleaching threshold

This isn't a useful number unless put into context.

Removed from conclusion

425: During January-March when ocean temperatures are warmest, years with mass bleaching have on average nine more doldrums days than non-bleaching years

From Fig 8b I see a difference of 3 or 4, Im not sure where the 9 comes from, and you haven't demonstrated if there is a statistically significant difference.

The conclusion statements were adjusted to reflect the statistical significance of these differences and we have removed the mention of specifically 9 more days. This number comes from comparing the means of the bleaching and non-bleaching years for the Dec-Apr period (Fig 8a).

432: The most prominent mechanism in our analysis is Rossby-wave breaking

'most prominent' implies that you looked at other mechanisms

Revised to 'This could be achieved by Rossby-wave breaking...'

Citation: https://doi.org/10.5194/egusphere-2025-3639-RC1

Response to reviewer #2

Main recommendation:

This paper seeks an explanation for coral bleaching events in certain weather/synoptic regimes and uses a K-Clustering technique to find those regimes. It addresses the character of those regimes, and links them to statistics on bleaching. The paper is overall well-written and clear, yet very descriptive and lengthier than need be to highlight the main findings. My recommendation would be to considerably shorten by drawing out what really matters. While some material is nice-to-have, it might better go into an appendix or supplementary material.

The paper is for example lengthy on the transitions between clusters, which relates to larger-scale dynamics and feedbacks between the ITCZ, tropical deep convection and the surface, which, while intriguing, is not the key focus of this paper or would require substantially different analyses. The paper ends with a conceptual figure 10, which is appreciated, but how clusters transition is not key to the bleaching events per se, and figure 10 tries to explain how they occur sequentially.

What seems a key results, but which gets lost in the text, is that sustained days in the non-trade clusters – as well as a lack of early season cooling – have a relation to CBEs. An autocorrelation or persistence statistic is not presented.

I also imagine that what is important is how LHFs progress during the day (along with cloudiness), which is not analyzed. The diurnal cycle in SST and heat fluxes can be substantial in the doldrums, but the paper looks at daily mean fluxes. In other words, the # of sustained days in the non-trade cluster and the minimum LHFS/cooling rates observed during those days seem important to understand CBEs.

The conclusions are short and nice. The target is thus to do a major revision on the rest of the paper to bring out the essentials by removing material that is nice but not key, and to look at the diurnality of fluxes and heating/cooling rates. I describe these points more specifically below.

We have shortened and restructured many sections to better highlight the key points and remove repetition.

We agree that the cluster durations are an interesting component of this analysis and have included a new table in section 3.1 to highlight the cluster persistence differences as well as a new figure in section 4 to compare cluster persistence in bleaching and non-bleaching years.

We agree that diurnal cycles are an important component of the meteorology in this region. However this is currently outside the scope of our manuscript as it would provide additional complications in the initial cluster analysis.

• Introduction: your objective is in the last paragraph; it would be nice to turn this into a more specific question. What are you specifically interested in? E.g. are all CBE's preceded by a trade-wind breakdown?

We have rephased our objectives into two questions in the revised manuscript

1. What weather regimes are conducive to heating and cooling on the GBR?

- 2. Are all recent GBR CBEs preceded by a trade wind breakdown?
  - L52: boundary layer and trade-wind layer in literature are often used to denote the same thing -.

The manuscript has now been revised to only 'trade-wind layer'

• L24; the doldrums are areas with subsidence, which would brin down drier air. Why is the humidity high? Or perhaps, do you want to say the humidity is confined to layers close to the surface (in the presence of subsidence).

Changed to high surface humidity

L25: boundary layer turbulence – do you mean convection?

Yes, we have adjusted this line to convection.

• Figure 1: what does N represent? Number of days in each cluster? I don't think the latter, because the number differs from what is shown as number of days in a later figure. Can you denote its meaning in the caption?

We have added clarification in the caption of both figure 1 and figure 2 that N represents days in each cluster. The numbers of days in figure 5 and table 3 (relative humidity and rain accumulation) are different as only select years were available under these data sets. This is now noted in their respective captions too.

Section 2.1. A large diurnal cycle in surface heat flux seems to be present over the
doldrums from work I have seen. What is your motivation for picking the specific
time of 10 LT? Is it a better constrained analysis? Are you avoiding certain sea
breeze type of circulations? And why do you not pick the peak cooling/heating rates
that are present in a day?

Using a daily average would create complications with diurnal cycles. We chose 0000 UTC as it coincides with the release time of the neighbouring observational sounding at Townsville and Willis Island. As these soundings would be ingested into the reanalysis dataset, it should be when the ERA5 data is closest to the observations. This explanation has now been added to the revised manuscript at line 69.

Using just the 0000 UTC time step also allows us to analyse the cloud cover and air mass origins for this single time step, a process that would again be complicated by diurnal cycles if we took a daily average.

Similarly, the timing is useful for comparing our work with other studies of the trade-inversion layer which use real soundings (e.g. Pope et al. (2009) and Lyons and Bonell (1992) use 2300 UTC).

Pope, M., Jakob, C., and Reeder, M. J.: Regimes of the North Australian Wet Season, Journal of Climate, 22, 6699–6715, https://doi.org/10.1175/2009jcli3057.1, 2009

Lyons, W. F. and Bonell, M.: Daily meso-scale rainfall in the tropical wet/dry climate of the Townsville area, north-east Queensland during the 1988–1989 wet season: Synoptic-scale airflow considerations, International Journal of Climatology, 12, 655–684, https://doi.org/10.1002/joc.3370120702, 1992

Trajectories: I am a bit confused about specific days in the northerly wind cluster, of
which many appear to have a back trajectory that goes towards the southeast/
Does this not tell us that the day-to-day weather variability/synoptic variability is
very strong and that subsequent days fall into one cluster or the other. Also, a
cluster does not tell us about persistent weather regimes per se, which seem key.

You're right that there is more variability in the two non-trade clusters and that not all days look as similar as the trade wind clusters and have added this point to the revised manuscript (line 172).

The clusters through their back trajectories do demonstrate the presence or absence of the trade winds which in this area is the dominant weather regime. We can also see the presence of the Australian Monsoon in the northerlies cluster being another prominent weather regime.

 Lines 144 – 145: I also see that in Fig 1 there is still that eastern high-pressure ridge, but additionally, the L pressure on the northeastern side of Australia is extended further down, which seems to me one of the main differences between the trades and non-trades. Also, in L376 you highlight this main difference (Low pressure area south of Davies Reef).

We have fixed this point in the revised manuscript L150-155

• In section 3.1 it appears that some statistical analysis such as autocorrelation would be helpful to address how long a certain cluster is sustained. it would be very interesting to learn about the persistence of one cluster, as it seems that persistence of doldrum conditions has more important on bleaching than if doldrum days persist only shortly and alternate with other clusters. In that sense the number of days in one cluster alone is not telling us the full story. You address the persistance only later in Lines 280-297.

We have added a table 2 to section 3.1 to include some statistics on cluster durations.

We have included a new figure on persistence where we compare the cluster durations between bleaching and non-bleaching years. As the total days in each cluster are different and there will obviously be more samples in the non-bleaching subsets, each group is divided by the total days in the group (i.e. the total days in the classic trades in bleaching years) to normalise the data.

**Cluster durations**

|            | Classic Trades | Summer Trades | Wet Trades | Doldrums | Northerlies |
|------------|----------------|---------------|------------|----------|-------------|
| 1-2 days   | 102            | 183           | 228        | 155      | 109         |
|            | (17.4%)        | (23.9%)       | (29.8%)    | (27.1%)  | (28.9%)     |
| 3-5 days   | 50             | 111           | 111        | 78       | 59          |
|            | (26.1%)        | (40.6%)       | (38.8%)    | (34.6%)  | (40.4%)     |
| 6-10 days  | 30             | 38            | 34         | 40       | 16          |
|            | (30.4%)        | (26%)         | (23.1%)    | (33.7%)  | (21.2%)     |
| 11-20 days | 8              | 7             | 7          | 3        | 4           |
|            | (14.1%)        | (9.3%)        | (8.3%)     | (4.7%)   | (8.9%)      |
| >20 days   | 3              | 0             | 0          | 0        | 0           |
|            | (12%)          |               |            |          |             |

Table 2. Cluster duration event counts with the percentage of days captured within each duration period.

 I find the transition table to be far less interesting than statistics on the number of persistent days during which non-trade events exist.

The transitions section has been shortened, however the transition table is important in highlighting how to move in and out of the trades. As we identify the wet trades as the main transition pathway, it also helps connect why this cluster has the strongest cooling despite not having prominent cool SSTAs.

• I find Figure 10 and lines 391 – 405 to be very descriptive (with Fig 10 complex) and not key to explaining bleaching events, which are not uniquely related to a transition between clusters. In some sense explaining the large-dynamics of the trades itself goes beyond the scope of this work.

The aim of this paragraph is to highlight how the transitions could be used to enhance predictability of these weather types which is essential for understanding how long these thermal stress periods last as the transitions and persistence are ultimately connected.

 Section 3.2 is overall very descriptive and detailed, so that key information is somewhat hidden, such as that the trades regimes are cooling SST in the GBR region, while the doldrums/northerlies regimes are warming the SST. Could this section better highlight the key information needed to understand the CBE events? Can other information/details go into an appendix?

We have shortened this section from 533 to 470 words and restructured it to better highlight the key information, e.g. heating and cooling over the reef and how that connects the synoptics

• Section 3.3, while nice, could also be shortened. There are repeating sentences.

We have shortened this section from 454 words to 370

 Section 3.4 MJO could also be shortly summarized by stating that between trades and non-trade clusters you transition from inactive/suppressed to active phases of the MJO.

This section has been shortened from 358 to 255 words

In section 5 - last paragraph running from page 19 – 20, there is notable repetition in results discussed, except for the important point of duration and persistence of the doldrums, which I would highlight as one of the key aspects that should not come at the very end of the results hidden in between a paragraph. I would encourage to make this analysis on the persistence stronger.

To support the persistence statistics in section 5, we have added a general cluster persistence statistics table 2 to section 3.1 and persistence distributions in section 4. More reference to the durations has been added to the discussion and conclusion sections.

Figure 9. Cluster durations split the 1996-2024 data into mass bleaching ('98, '02, '06, '16, '17, '20, '22, '24) and non-bleaching years. As the total days in each cluster are different, each bleaching and non-beaching duration frequency is normalised by dividing through the amount of days in the respective group.

In the Discussion/section 6 (1st paragraph) it is discussed that CBEs relate to a
persistence in non-trade clusters. Given that those seem particularly important,
would it be good to look at what Pacific-basin wide synoptics were present during

these persistent periods in particular? (or do you think these are well reflected in the cluster anomalies in Fig 4.

We are currently working on a follow-up manuscript that examines what drives persistence in mainly the doldrums cluster. From our early analysis, the mean cluster anomalies do represent the general basin wide synoptics for these long duration periods.

**Grammatical errors:**

L69: 'was found to' - > 'were found to' corrected

L85: for each day in each cluster or each cluster? corrected

L294: respectively instead of respectfully? corrected

L 372: 'to form creating local SST heating' incorrect grammar corrected

L 386: fix citation style (Windmiller, 2024) - > Windmiller (2024) corrected

Citation: https://doi.org/10.5194/egusphere-2025-3639-RC2